# ON THE IMPORTANCE OF LOOKING AT THE MANIFOLD

## ABSTRACT

Data rarely lies on uniquely Euclidean spaces. Even data typically represented in regular domains, such as images, can have a higher level of relational information, either between data samples or even relations within samples, e.g., how the objects in an image are linked. With this perspective our data points can be enriched by explicitly accounting for this connectivity and analyzing them as a graph. Herein, we analyze various approaches for unsupervised representation learning and investigate the importance of considering topological information and its impact when learning representations. We explore a spectrum of models, ranging from uniquely learning representations based on the isolated features of the nodes (focusing on Variational Autoencoders), to uniquely learning representations based on the topology (using node2vec) passing through models that integrate both node features and topological information in a hybrid fashion. For the latter we use Graph Neural Networks, precisely Deep Graph Infomax (DGI), and an extension of the typical formulation of the VAE where the topological structure is accounted for via an explicit regularization of the loss (Graph-Regularized VAEs, introduced in this work). To extensively investigate these methodologies, we consider a wide variety of data types: synthetic data point clouds, MNIST, citation networks, and chemical reactions. We show that each of the representations learned by these models may have critical importance for further downstream tasks, and that accounting for the topological features can greatly improve the modeling capabilities for certain problems. We further provide a framework to analyze these, and future models under different scenarios and types of data.

## 1 INTRODUCTION

The ability to recognize relational information between or even within individual percepts is one of the fundamental differences between human and artificial learning systems. For example, the feature-binding problem (Roskies, 1999), i.e. the mechanism governing the visual system to represent hierarchical relationships between features in an image, is still largely unsolved by neuroscientists, exacerbating the development of bio-inspired statistical learning systems. Traditional relational learning approaches mostly sort into either learning internal or external relational structure between samples and rely heavily on crafting domain-specific expert knowledge that is engineered into the model (Struyf & Blockeel, 2010). Consequently, these models have yet to prove their usability in real applications and, although some neurocomputational frameworks for relational learning were proposed (Isbister et al., 2018), building statistical models that explore higher-order dependencies between samples remains a key challenge for computer vision and robotics application. Consequently, relational reasoning has been advocated a pivotal role for the future of artificial intelligence (Battaglia et al., 2018). On the very contrary, deep learning as a purely data-driven approach has enjoyed remarkable success in recent years by learning complex non-linear functions mapping raw inputs to outputs without explicit dependency modelling. Fields like relational reinforcement learning (Džeroski et al., 2001) and statistical relational learning (Koller et al., 2007) aimed to fill this gap; but recently augmenting deep (reinforcement) learning models toward relational reasoning emerged as a promising approach (Zambaldi et al., 2018; Zhang et al., 2016). Many successful contributions for relational modelling in images however largely rely on Euclidean spaces (Dai et al., 2017; Yao et al., 2018).

It is widely agreed that graphs are the ideal structure to enable relational deep learning (Hamilton et al., 2017). Prior work has shown that metagraphs incorporating relational information about the

dataset can improve unsupervised representation learning in finding less complex models that preserve relational information without loosing expressivity on the original representation (Dumancic & Blockeel, 2017). In terms of predictive modelling, the relational representations can be superior to ordinary ones (Dumancic & Blockeel, 2017) and graph-induced kernels can aid in improving phenotype prediction compared to non-topological kernels (Manica et al., 2019). In generative modelling, relational distribution comparison was demonstrated to facilitate the learning of generative models across incomparable spaces (Bunne et al., 2019).

Here, we perform an extensive study on the impact of the topological information in learning data representations. Specifically, we focus on the trade-off between leveraging data point features and relational information. We consider a selection of unsupervised models for learning representations lying in different areas of the spectrum. Ranging from Variational Autoencoders (VAEs) (Kingma & Welling, 2013) to node embedding techniques based on random walks on graphs (Grover & Leskovec, 2016), passing through graph neural networks (Veličković et al., 2018) and the proposed Graph-Regularized Variational Autoencoders (GR-VAE), our adaptation of VAEs where the latent space is regularized through a metagraph representing relations between samples of the dataset that we introduce in this work.

The methods considered are evaluated on different datasets and downstream tasks where the impact of the topology can be appropriately assessed. Initially, we examine the impact of implicitly accounting for the topology to validate the GR-VAE in two synthetic studies based on topologically connected 4D point clouds and MNIST (LeCun et al., 2010) with added relational information based on the labels. After this initial validation, we move to evaluating all the methods in the case of text representations and chemical reactions. In the case of text representations, we analyze the methods performance on Cora, CiteSeer, and PubMed (Sen et al., 2008), using their citation networks, and evaluating the representations learned in a downstream classification task. Finally, we study the impact of the topology in molecule representations using a chemical reaction dataset (Jin et al., 2017), where the downstream task consists in predicting the reactivity of reactant/reagent-product pairs.

## 2 METHODS

In this section we present the different models compared in this study. Our approach is to explore a spectrum of models with varying availability of features and topology (see Figure 1).

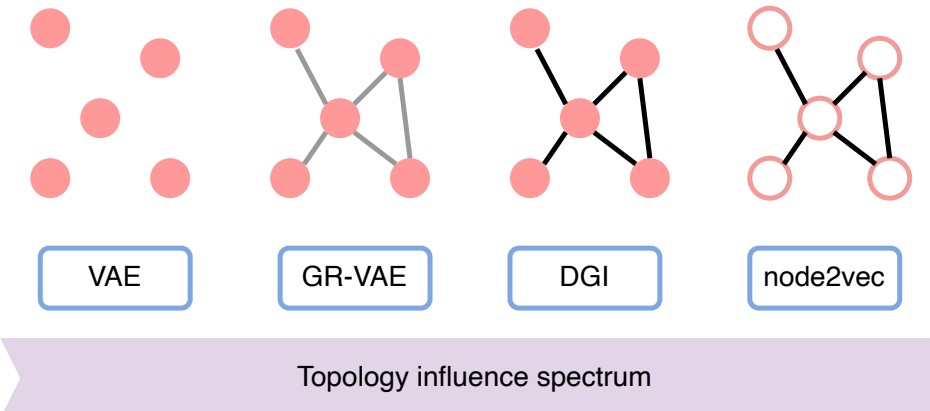

Figure 1: **Topology influence spectrum in the light of the model considered.** From left to right we selected the models in order to smoothly transition from a case where only the point/node features are relevant (left, standard VAE) to the opposite end of the spectrum where only the topological properties are considered (right, node2vec). In the middle we find the cases where the point node features and the topology are blended, either implicitly via a regularizer in the GR-VAE case or explicitly in the DGI case.

## 2.1 IMPLICIT TOPOLOGICAL LEARNING

We first explore VAEs (Kingma & Welling, 2013) which only intake features from the nodes, thus serving as a baseline model agnostic to topological information.

**Graph Regularized VAE.** We then introduce a variation of VAEs (Kingma & Welling, 2013), defined as Graph-Regularized VAEs (GR-VAE), that augments the trade-off between reconstruction error and KL divergence by a topological constraint (see Figure 2 for an overview). In GR-VAE, the latent space is regularized through a metagraph present in the data. We suggest that accounting for available information on how different samples relate (henceforth simply referred as the graph) may help at obtaining more powerful representations, especially where this relationships are directly involved with a downstream task of interest.

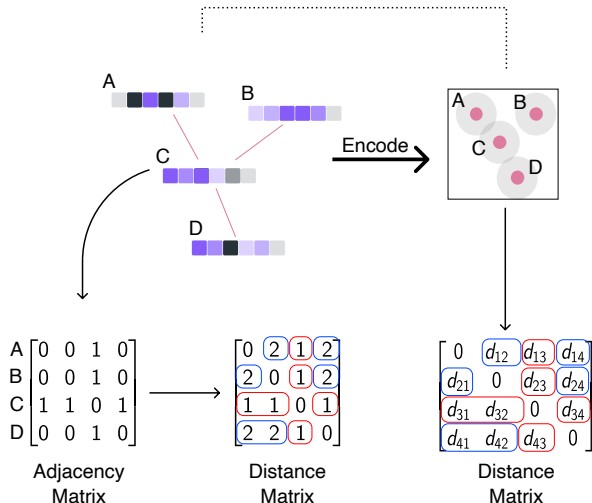

Figure 2: **Definition of the graph used to build the regularization term in the GR-VAE formulation.** Relational information between the data points (A-D) is considered to define a dataset metagraph that can be used to constraint distances in the learned latent space. The notation $d_{\nu i}$ in the distance matrix abbreviates $d_D(\nu, i)$ from Equation 1.

Our approach adds a term to the loss defined by the set of constraints to the samples' representation in the latent space given the distances of the samples' metagraph. For a given set of samples, $\mathbb{S}$, we can compute their distances in the latent space D, as well as over the graph $\mathcal{G}$. For each node, $\nu$, we expect the distances to the other nodes, once embedded in $D$, to resemble the distances over $\mathcal{G}$. Thus, we enforce a constraint aimed to preserve the relative distances in the two spaces. Formally, fixing a node $\nu$ and considering any pair of nodes $(i, j)$, we can define the following penalty term:

$$\phi(d_D, d_{\mathcal{G}}, \nu, i, j) = \begin{cases} (d_D(\nu, j) - d_D(\nu, i))^+ & \text{if } d_{\mathcal{G}}(\nu, i) > d_{\mathcal{G}}(\nu, j) \\ (d_D(\nu, i) - d_D(\nu, j))^2 & \text{if } d_{\mathcal{G}}(\nu, i) = d_{\mathcal{G}}(\nu, j) \\ (d_D(\nu, i) - d_D(\nu, j))^+ & \text{if } d_{\mathcal{G}}(\nu, i) < d_{\mathcal{G}}(\nu, j) \end{cases} \quad (1)$$

where $d_D$ and $d_{\mathcal{G}}$ are metrics defined in the latent space and over the graph respectively. In the following, we select the Euclidean distance as $d_D$ and the geodesic distance (Floyd, 1962) as $d_{\mathcal{G}}$. The overall loss function of the GR-VAE thus becomes:

$$\mathcal{L}_{\text{GR-VAE}}(\boldsymbol{X}; \boldsymbol{\theta}) = \mathcal{L}_{\text{VAE}}(\boldsymbol{X}; \boldsymbol{\theta}) + \gamma \sum_{\nu \in \mathbb{S}} \sum_{(i,j) \in \mathbb{S} \times \mathbb{S}} \phi(d_D, d_{\mathcal{G}}, \nu, i, j) \quad (2)$$

where $\boldsymbol{X}$ are the features of the samples in $\mathbb{S}$, $\boldsymbol{\theta}$ the network parameters and $\gamma \geq 0$ regulates the strength of the penalty.

## 2.2 Explicit topological learning

Notably, GR-VAE is devised to infer topological information solely from a soft constraint, without any architectural requirements such as graph convolutions. On the other side of the spectrum, graph neural networks (GNN) instead model topology *explicitly*.

**Deep Graph Infomax.**  Here, we consider to a Deep Graph Infomax (DGI), a state-of-the-art GNN for unsupervised representation learning (Veličković et al., 2018). DGI relies on maximizing mutual information between subgraphs (themselves derived with GCNs) yielding representations that facilitate downstream node-wise classification tasks. These two models used as a comparison for the text classification task.

**GAE and VGAE.**  We consider both non-probabilistic Graph Autoencoders (GAE) and Variational Graph Autoencoders (VGAE) (Kipf & Welling, 2016) which are models for unsupervised-graph data inspired on the VAE.

**GraphSAGE.**  We consider GraphSAGE, an inductive framework that generates node embeddings by sampling and aggregating the features for the local neighborhood of a node (Hamilton et al., 2018). GraphSAGE is used as a comparison model for the text classification task.

**node2vec.**  Finally, we utilize node2vec (Grover & Leskovec, 2016), which only consumes topological information but no node-specific features. The node2vec algorithm learns a compressed feature space that maximizes the probability to preserve local neighborhoods. With the exception of node2vec, the specific details for the configuration of each model will depend on the dataset we are evaluating on, thus will be detailed in each of the datasets' results.

## 2.3 Datasets

In the following we describe the datasets used and the experimental setup for the downstream tasks.

### 2.3.1 Synthetic data: a qualitative assessment

First, we consider a synthetic dataset with arbitrarily generated graphs on a plane. Each node's features will be composed by the combination of the first two edges directions' (in the case of nodes with a single edge the feature vector is padded with zeros), resulting in a feature vector of 4 dimensions. Thus, each node holds partial, yet insufficient topological information about the graph. As described above, the entire graph is then used to regularize the latent space.

### 2.3.2 MNIST

On a similar line we expanded this experiments by taking MNIST (LeCun et al., 2010) and generating a topology across the different labels by chaining the samples from 0 all the way to 9. We use this dataset to further test the model's capability of affecting the topology of the latent where the individual node features are of higher complexity, at least when compared to the synthetic data, while maintaining comparable reconstruction performance to the non-constrained scenario.

### 2.3.3 Text representation

We evaluate three classification datasets: Cora, CiteSeer, and PubMed (Sen et al., 2008). These datasets contain networks of documents linked by the citation links between documents. The text of the document is represented as a bag-of-words, which we take as a feature vector for each of the documents. Furthermore, each document corresponds to a particular task. We divide each dataset, and use a part of it to train the embedding and the other part on a downstream class prediction task, using the embedding model mentioned above.

### 2.3.4 Chemical reaction representation

Finally, we analyze the influence of the topology in learning effective representations for molecules in the context of chemical reactions, a topic that has testified a surge in popularity in the recent past as

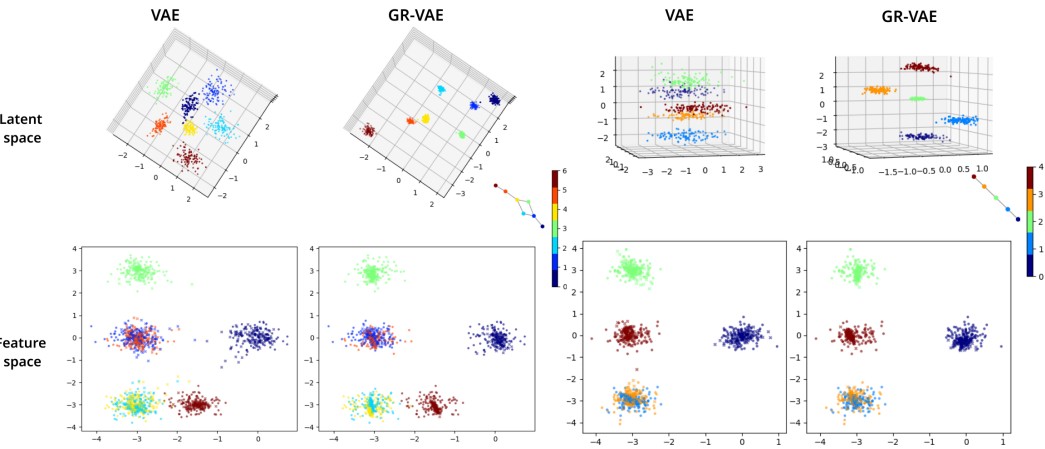

Figure 3: **Learned representations on two synthetic datasets with diferent topology.** This figure displays qualitatively how GR-VAE affects the latent space topology under different conditions, specifically when compared to its non-constrained counterpart. The plots on top show the latent space, the ones in the bottom show the feature space, for the first two features. The two datasets (left and right) had different topologies, shown as a graph, next the color bar (all colors across plots correspond to the nodes' ids). On the features plots both the original data (round marker) and the reconstructed data (cross marker) are shown.

a field for deep learning applications (Schwaller et al., 2019; 2020). To this end, we adopt the dataset compiled by Jin et al. (2017) where we represent reagents, reactants and products using SMILES representations (Weininger, 1988), using the splits provided. For each molecule we extract features using the encoder of a VAE based on stack-augmented GRU layers (Joulin & Mikolov, 2015), as proposed in Born et al. (2020), pretrained on PubChem (Kim et al., 2015) (more details can be found in the Appendix A.2). As for the topological reaction representation we consider a bipartite graph connecting the products to all the reactants and reagents. Each reaction bipartite graph is then used to generate the resulting final graph connecting all the nodes that are shared between different reactions. Using the training split provided by Jin et al. (2017), the models are finetuned as follows: VAE at molecule level, GR-VAE and DGI at reaction level (GR-VAE in an implicit form through the loss regularizer), node2vec on the aggregated graph. Furthermore, DGI uses the different VAEs and GR-VAEs as part of its encoder.

To evaluate the quality of the representations learned and the impact of the topology, we consider the task of predicting whether two molecules are respectively reactant/reagent and products of a valid chemical reaction. The resulting binary classification task has an inherent relation with the underlying reaction network. For VAE, GR-VAE and node2vec we represent a pair of molecules as the concatenation of the encoded molecules/nodes in the respective latent spaces. In the DGI case, we represent the pair as the embedding of a graph connecting the molecules. These representations are then trained on the validation split and later evaluated on the test split as defined by Jin et al. (2017).

## 3    RESULTS

We break our experimental results in two parts based on the division previously made (in Section 2). First, we analyze the validity of implicitly learning the topology through the proposed extended VAE formulation. Secondly, we compare all the different modalities, including implicit and explicit topology on a set of different task.

Table 1: **Quantitative results for the MNIST experiment.** We report results for three different models with varying number of dimensions in the latent space: 3, 16, and 64. For each one we explore four training setups, a regular VAE ($\gamma = 0$) and three intensities of GR-VAE ($\gamma \in \{1, 10, 100\}$). We then report the reconstruction loss, the silhouette score of the test samples in the latent space, and the Hamiltonian. Furthermore we train two downstream model: k-NN and a classification tree, and we report their average F1 scores over a 5-fold cross-validation.

| Latent dimensions | 3 | | | | 16 | | | | 64 | | | |
|---|---|---|---|---|---|---|---|---|---|---|---|---|
| GR $\gamma$ | **0** | **1** | **10** | **100** | **0** | **1** | **10** | **100** | **0** | **1** | **10** | **100** |
| Reconstruction loss | **141** | 143 | 147 | 172 | 83 | **82** | 84 | 105 | 81 | **79** | 81 | 95 |
| Silhouette score | .052 | .092 | .216 | **.195** | .074 | .096 | .141 | **.178** | .055 | .060 | .112 | **.168** |
| K-NN | .634 | .711 | .766 | **.816** | .928 | .933 | **.946** | .940 | .938 | .941 | **.947** | .937 |
| Tree | .574 | .643 | .700 | **.753** | .737 | .728 | .808 | **.818** | .692 | .725 | .777 | **.819** |

## 3.1 A VALIDATION ON THE IMPLICIT TOPOLOGICAL LEARNING

In this section we present the results for the described experiments on implicit topological learning using VAE and GR-VAE.

### 3.1.1 SYNTHETIC DATA

We perform different analysis to better understand how the addition of topological information may influence the representations that a model will learn. For that we generate a synthetic dataset (described in Section 2.3.1) and train both a regular VAE and a GR-VAE. Qualitatively the results of this experiment (Figure 3) show us the stark influence that accounting for the structure has in the final learnt embedding. We can see that, while both models are able to achieve a comparable reconstruction, the latent space in the GR-VAE still preserves the topology of the metagraph in the data, and one is able to reconstruct this from the different clusters in the latent space. This simple experiment already shows how one may achieve scenarios where the metagraph topology does in fact determine the latent space topology, and motivates further the rest of the experiments in this study.

### 3.1.2 MNIST

Here we discuss the set of experiments conducted on the MNIST dataset where we impose a synthetic topological structure in the form of a chain following the numerical ordering of the labels (from 0 to 9). In Table 1 we show how the model performance, in terms of reconstruction error, Silhouette score (Rousseeuw, 1987) and F1 score in a downstream task (i.e. label prediction) are affected by changing the latent dimensions and the weight of the graph regularizer. Interestingly, we can see how the topological information, even if in the form of a simple chain, is helping in improving the quality of the learned image representations. Qualitatively, we can also observe in Figure 4 how the latent space can benefit from including topological information indirectly regularizing the loss, that becomes quite evident when looking at the distance matrix of the of the centroids.

## 3.2 EXPERIMENTS ON THE FULL TOPOLOGICAL SPECTRUM

In this section we present the results for the described experiments on explicit topological learning using all the models introduced in the the methods (Section 2).

## 3.3 TEXT REPRESENTATIONS

For the text datasets (Section 2.3.3) we adapted the models such that it would be fitting with the data and structure. For the VAE we decided to use a relatively simple model composed of two fully connected layers with ReLU activation, for both the encoder and the decoder. For DGI we followed the setup described in the original paper (Veličković et al., 2018), where the encoder is composed by a one-layer Graph Convolutional Network (GCN) (Kipf & Welling, 2017) with PReLu (He et al., 2015) activation. On all the models, the learned representations was then used to train a logistic

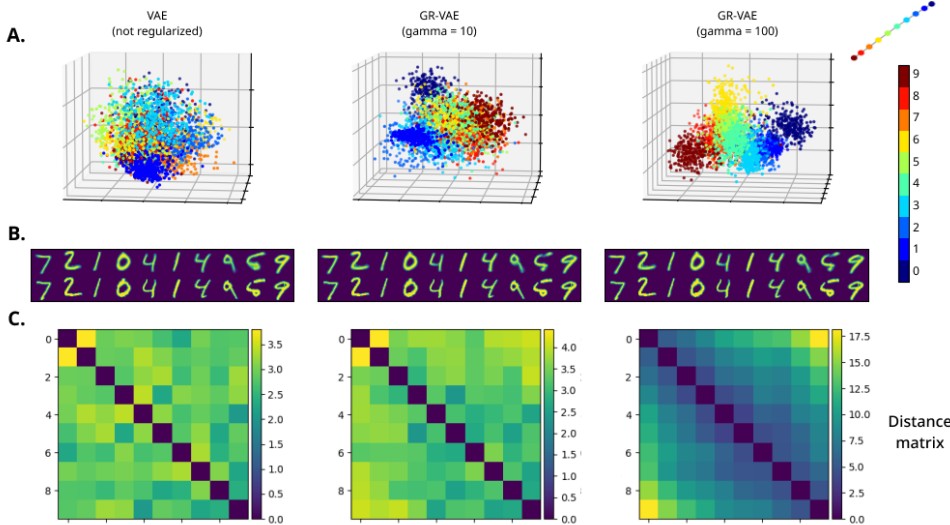

Figure 4: **Qualitative analysis of the latent representations learned in the MNIST case. A.** PCA projection of the samples in the latent space under different training regimes. The original latent space has 16 dimensions. The metagraph is a chain connecting each class from 0 to 9 in order (a representation can be seen on the top right). The samples can be seen coloured by class pertinence. **B.** Display of a reduced set of the original samples (bottom row) and their reconstructions (top row). **C.** Distance matrix between the centroids of each label's point cloud.

regression on a set of train nodes and predict each document's (i.e. node) class. We, then, report the prediction accuracy on the test nodes. The splits were reused from Yang et al. (2016).

Furthermore, in order to be able to train the GR-VAE we create batches by sub-sampling the graph around each of the nodes in the graph , an instance of the DGI model was also trained under the same circumstances in order to fairly compare both setups, where to be the case that the sub-sampling affected the performance. The results for the text task can be seen in Table 2.

In order to train the GR-VAE we create batches by sub-sampling the graph around each of the nodes in the graph. In order to generate each of these batches, for each node in the graph we sample 20 nodes in its neighborhood (up to 2 hops away from the node). This setup, where at a particular step the model only has access to a partial observation of the entire graph, is also applied to DGI as a separate setup. Furthermore, we explore, under this condition, how pre-encoding the node information (using either VAE or GR-VAE) affects the performance of DGI when compared to using the raw node features directly. The results for this experimental setup can be seen in Table 3.

Our results show clearly the strong performance of DGI in the three datasets. Interestingly DGI's performance drops when only obtaining batched information. As the authors point out when comparing to GCN, DGI seems to benefit from the fact that it has access to the entire graph (Veličković et al., 2018). Node2vec outperforms both VAE and GR-VAE in Cora and PubMed, however it falls behind in CiteSeer. We assume that that difference arises due to the relative importance of the graph topology in the different datasets. In CiteSeer, it is likely that, features alone hold more information of the underlying class than the citation and references network around each document. The relationship between VAE and GR-VAE also reflects this balance, whereas in Cora the addition of topological information seems to aid to the structuring of the latent space and help in creating a more useful representation, the oposite happens in CiteSeer. That duality shows how this information may aid in cases where it's more relevant for the downstream task, but it may hinder in cases where the direct link between topology and class (or downstream task) is weaker or straight

Table 2: **Results on the text representations.** Accuracy results for the text classification task in the Cora, CiteSeer, and PubMed dataset. In this particular experiment GR-VAE model was trained with equally weighted factors ($\gamma = 1$) of the loss components (reconstruction, KL-divergence and graph regularization).

| Model | Cora | CiteSeer | PubMed | Input data |
|---|---|---|---|---|
| Random | 0.152 | 0.152 | 0.322 | – |
| VAE | 0.530 | **0.531** | **0.525** | V |
| GR-VAE | **0.607** | 0.492 | 0.32 | V, E |
| DGI | **0.819** | **0.684** | 0.736 | V, E |
| GAE | 0.779 | 0.644 | 0.774 | V, E |
| VGAE | 0.759 | 0.569 | 0.718 | V, E |
| GraphSAGE | 0.718 | 0.538 | **0.765** | V, E |
| node2vec | 0.719 | 0.464 | 0.676 | E |

Table 3: **Results on text representations with partial graph observability.** Accuracy results for the text classification task in the Cora, CiteSeer, and PubMed dataset. In this particular experiment DGI was trained using the same batching procedure as GR-VAE model was trained with different values for the graph regularizer weight, $\gamma$.

| Model | GR $\gamma$ | Cora Latent space size 20 | 100 | 250 | CiteSeer 20 | 100 | 250 | PubMed 20 | 100 | 250 |
|---|---|---|---|---|---|---|---|---|---|---|
| **DGI (VAE)** | - | 0.738 | 0.723 | 0.746 | 0.611 | **0.650** | 0.635 | 0.457 | 0.558 | **0.620** |
| **DGI (GR-VAE)** | 0.5 | 0.701 | 0.732 | 0.744 | **0.663** | 0.616 | 0.612 | 0.384 | 0.418 | 0.448 |
| **DGI (GR-VAE)** | 1 | 0.73 | 0.756 | **0.761** | 0.612 | **0.644** | **0.646** | 0.429 | 0.431 | 0.410 |
| **DGI (GR-VAE)** | 2 | 0.705 | 0.734 | **0.764** | 0.624 | 0.626 | 0.630 | 0.522 | 0.477 | 0.312 |
| **DGI (GR-VAE)** | 10 | 0.527 | 0.7 | 0.576 | 0.597 | 0.635 | **0.654** | **0.594** | 0.499 | 0.377 |
| **DGI (GR-VAE)** | 100 | 0.326 | 0.49 | 0.361 | 0.301 | 0.426 | 0.355 | 0.442 | 0.541 | 0.524 |
| **DGI** | | | 0.738 | | | 0.611 | | | **0.722** | |

non-existent. Ultimately, this regularizer can lower signal-to-noise ratio in cases where a particular topology is irrelevant.

The results (Table 3) on with partial observability reflect the same observations stated previously. DGI performs worse than in the previous setup (Table 2), which was an expected result. We see that under some circumstances pre-encoding the node features with either VAE or GR-VAE may improve the performance. In particular the observation about the relationship between VAE and GR-VAE (made in the previous results) is also reflected in these results. In spite of this, the results appear to show high variation and not a clear pattern of what parameter combination increases performance. A reason for that could be the unaccounted randomness introduced by the sampling method used in the batching procedure, which may provoke variation in accuracy from model to model.

## 3.4 CHEMICAL REACTIONS

The results of the experiments run on the chemical reactions dataset can be seen in Table 4. Similarly to the text dataset, using DGI gave the best performance. Although the results are more nuanced, ultimately the encoder used was critical. For instance, when using an encoder pretrained in a different dataset we could find situations where the DGI performs worse than the VAE encoder alone. The opposite end is when combining all the methods used in the study, where DGI using node embeddings finetuned with a GR-VAE achieves the highest accuracy. It is extremely interesting to observe such a behavior, where we see that among the three top performing models a plain VAE with no topological information is present. This seems to suggest that the quality of the SMILES embedding is key in the task considered.

Table 4: **Results for chemical reactions experiment.** We report the accuracy on the downstream reaction task. The annotations in parenthesis specify details about the encoder: *Finetuned* denotes that the VAE or GR-VAE has been finetuned on chemical reaction data (on a different split from the downstream reactions), in the case of the DGI the annotation references to which VAE model was used for encoding the SMILES. For each instance of the GR-VAE we display which $\gamma$ we used in training.

| Model | GR $\gamma$ | Accuracy |
|---|---|---|
| Random | – | 0.5 |
| **VAE** | – | **0.5740** |
| VAE (finetuned) | – | 0.5613 |
| GR-VAE (finetuned) | 0.5 | 0.5631 |
| GR-VAE (finetuned) | 1 | 0.5470 |
| GR-VAE (finetuned) | 2 | 0.5624 |
| GR-VAE (finetuned) | 5 | 0.5543 |
| DGI (VAE) | – | 0.5003 |
| **DGI (VAE finetuned)** | – | **0.6248** |
| **DGI (GR-VAE finetuned)** | **0.5** | **0.6602** |
| DGI (GR-VAE finetuned) | 1 | 0.5617 |
| DGI (GR-VAE finetuned) | 2 | 0.5507 |
| DGI (GR-VAE finetuned) | 5 | 0.5321 |
| node2vec | – | 0.5 |

## 4 DISCUSSION

In this study we explored the importance of topological information in learning data representations. We demonstrated the addition of inter-sample relational information as a means to improve these representations, and stressed the trade-off between leveraging data point features and relational information.

We have described a novel loss that expands the VAE by leveraging a relational metagraph and described under which circumstances this added factor becomes a support for further downstream tasks. Most evident are our MNIST results, where adding data that is directly linked to the downstream task of interest creates a more useful arrangement of the latent space, resulting in improvements of the downstream prediction using these embeddings in all the explored setups. It is worth emphasizing that the regularized introduced in the GR-VAE, can not only inject topological awareness into non-topological models, but also be combined with them to achieve superior performance in downstream prediction tasks—as we see in the chemical reaction case. Furthermore, we explore scenarios where the metagraph is less obviously linked to the end prediction. In those, the benefit of adding a graph regularizer (i.e. GR-VAE vs. VAE) is more subtle. Our work opens the door to further exploring ways to evaluate which representation are more useful for given downstream tasks as well as, creating metrics to quantitatively evaluate so. In short, this work aims to be a motivation for looking at the manifold and a small step towards understanding how inter-sample relational information can be beneficial, even in those cases where this data is not explicitly ingested by the model or where the link to a particular end goal may not be obvious.

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

## A  APPENDIX

### A.1  GRAPH REGULARIZED VAE DETAILS

In Figure 5 a detailed description of the GR-VAE architecture is depicted.

### A.2  SMILES EMBEDDING

The SMILES VAE used for the chemical reaction dataset was implemented following the description in (Born et al., 2020). It consists of two layers of stack-augmented GRUs (Joulin & Mikolov, 2015) in both encoder and decoder and is trained with teacher forcing (Williams & Zipser, 1989), token dropout (Bowman et al., 2015) and one-hot encodings.

The dataset consisted of 500,000 molecules represented as canonical SMILES strings from PubChem (Kim et al., 2015).

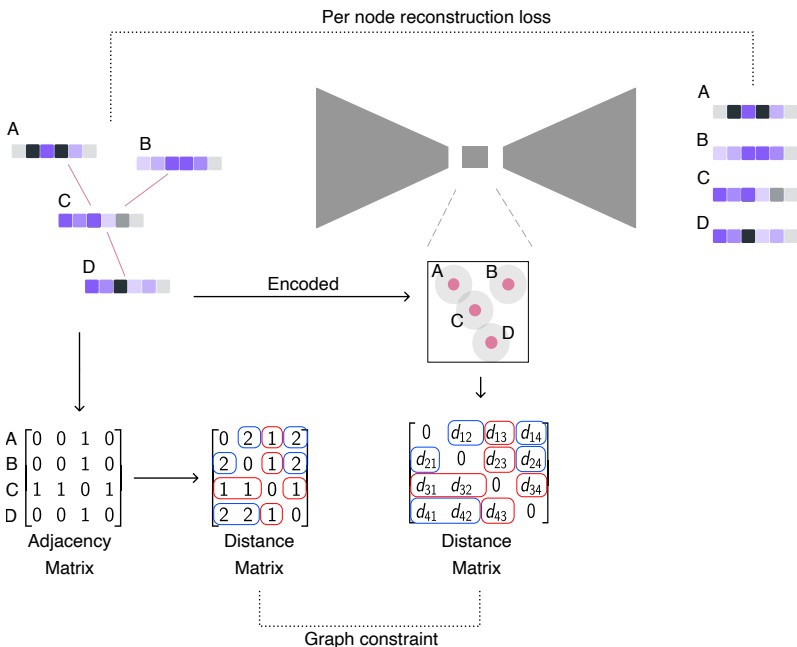

Figure 5: **Complete overview of the GR-VAE approach.** Notice that, the notation $d_{\nu i}$ in the distance matrix abbreviates $d_D(\nu, i)$ from Equation 1.

### A.3 EXTENDED IMPLICIT LEARNING

Here we show the extended results for the MNIST implicit learning tasks.

Figure 6 shows expanded results for a setup where the VAE was mapping to a latent space of 3 dimensions. In that case we can see that with a strong regularizer we still accomplish our desired objective of organizing the point clouds as a chain. This setup, with only 3 dimensions where to map the points, challenges the model and makes it more difficult to obtain reconstructions as faithful to the original images as those we saw with models with more dimensions (Figure 4). However it comes useful to display how the embeddings done using the graph regularizer can help at creating clear distinctions between sample groups. For instance, the non-regularized VAE mixes a number of digits (see Figure 6B), while the models that were regularized manage to reconstruct the same digit (i.e. class), usually at the expense of generating reconstructions that are less faithful to the original image in term of details or style

To validate the qualitative assessment on the model's ability to restore the original chain as shown in Figure 6C, we computed the Shortest Hamiltonian Paths (SHP) (Held & Karp, 1962) on a fully connected graph of 10 nodes (representing the centroids of the labels in the latent space) where the network topology (i.e. edge weights) was given by the pairwise distances of the centroids. If the SHP of such a graph is a chain from 0 to 9 it proves that the topology is preserved perfectly in the latent space. To compare the different chains we used Dynamic Time Warping Müller (2007), a distance measure based on time series alignment computed with `FastDTW`[1]. An optimal topology corresponds to a DTW distance of 0. The results for the later can be seen in Table 5, the full chains can be seen in Table 6.

---

[1]https://github.com/slaypni/fastdtw

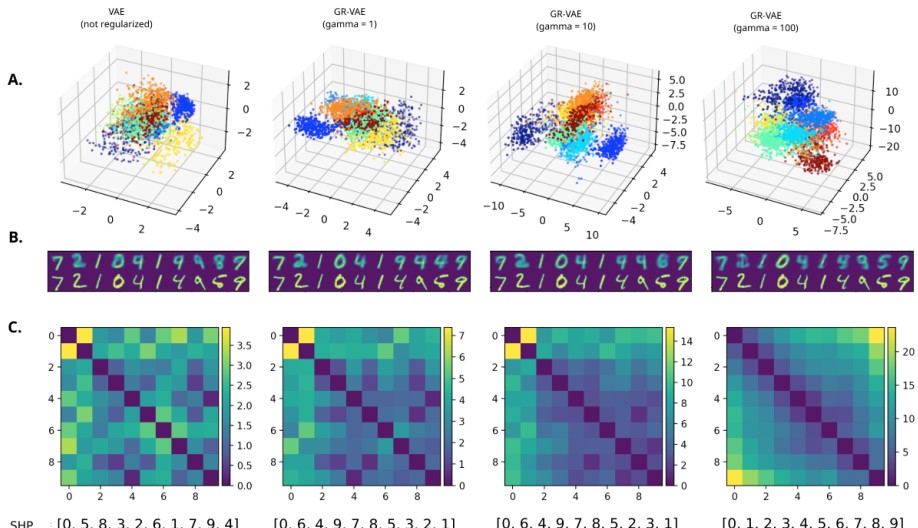

Figure 6: **Qualitative analysis of the latent representations learned in the MNIST case (with a latent space size 3).** Figure with extended information about the MNIST results, it displays same set of experiments run in Figure 4, but using a VAE with a latent space of 3 dimensions. For that reason **A.** directly displays all the latent space dimensions (not a PCA projection). It also includes an extra setting ($\gamma = 10$) for the regularizer.

Table 5: **Shortest Hamiltonian Path (SHP) distance to an ordered chain.** This table displays the distance from each SHP to an ordered chain (0 to 9) using FastDTW. For reference, the average distance of a random connected path is $30.03 \pm 6.5$ (computed with 1000 random sequences).

|  | Latent space dimensions | | |
|---|---|---|---|
|  | **3** | **16** | **64** |
| – | 23 | 23 | 27 |
| 1 | 31 | 31 | 27 |
| 10 | 0 | 31 | 3 |
| 100 | 0 | 0 | 0 |

Table 6: **Shortest Hamiltonian Paths**. Full chains obtained when running SHP over the class centroids of the samples in the latent space. We can see that with the biggest value of the regularizer ($\gamma = 100$) SHPs recover the original chain used for the constraint.

| $\gamma$ | Latent space dimensions | | |
|---|---|---|---|
|  | **3** | **16** | **64** |
| no regularizer | 0 5 8 3 2 6 1 7 9 4 | 0 6 2 8 5 3 1 7 9 4 | 0 6 4 9 7 1 3 5 8 2 |
| 1 | 0 6 4 9 7 8 5 3 2 1 | 0 6 4 7 9 8 5 3 2 1 | 0 6 4 9 7 1 3 5 8 2 |
| 10 | 0 6 4 9 7 8 5 2 3 1 | 0 1 2 3 4 5 6 7 8 9 | 0 1 2 3 4 5 6 8 9 7 |
| 100 | 0 1 2 3 4 5 6 7 8 9 | 0 1 2 3 4 5 6 7 8 9 | 0 1 2 3 4 5 6 7 8 9 |

