# OpenReview forum: "On the Importance of Looking at the Manifold"
_ICLR.cc/2021/Conference — Reject_

### Official Review · AnonReviewer3 · 2020-10-26
**The paper studies the importance of utilising manifold/topology information in prediction tasks. The method is not novel enough and the comparison seems problematic.**

**Rating:** 4
**Confidence:** 4

**Review:**

The paper focuses on studying the importance of utilising manifold/topology information for machine learning tasks. To this end, the authors benchmark four different approaches, including VAE, GR-VAE (using graph distances to regularise  embedding distances (as shown in Eq. 1)).  The paper performs experiments on four tasks, including synthetic data, MNIST, text representation, and chemical reactions. As conclusion, the paper demonstrates that in some cases, adding relational information is beneficial, while in other cases, the effect is subtle. Thus, the paper aims to provide a metric for understanding when and how manifold/topology information is needed.

Pros:

1. Instead explicitly learning a graph, this paper proposes an implicit method with graph regularisation.

2. The related work is well-explained. The paper did a well summarization of previous methods.

3. The paper performs extensive experiments to study the importance of manifold for prediction tasks.

Cons:

1. The latent graph method is not novel enough, since the method can be categorised as graph regularisation, which is a widely used method in recommendation and information retrieval. Could the author explain why this regularisation is picked from a spectrum of graph regularisation algorithms?

2. The comparison of the paper is problematic. First, the methods (DGI, node2vec, GR-VAE, VAE) compared are quite different methods. Can the authors confirm that the comparison is fair and meaningful (e.g. eliminating other confounding factors like controlling the number of parameters)? Second, I am not sure whether this comparison is optimal. In particular, to study the importance of relational information, other method can be used. For example, we can control adjacency matrix received by graph neural networks. We can totally ignore the edge information (like VAE in the paper) or use a predefined graph (e.g. a fully connected graph like in Transformer). In between, we can corrupt input graphs (e.g. randomly adding or deleting some edges) before feeding it to graph neural networks. This approach seems more reasonable to me for studying the importance of manifold. It is difficult to control these in this paper because the methods used in this paper are totally different  (e.g. DGI and GR-VAE differs in both loss function and input format). So, the conclusion of the paper is skeptical. It mainly justifies which method can perform better in downstream tasks instead of justifying the importance of manifold.

3. The introduction is lengthy and should be more focused on the contribution of this paper. Similarly, the other sections need a major revision to highlight the contribution, as the main contribution of the paper lies in the implicit graph regularisation and a comparison of a series of methods with/without relational information.

4. Some baseline methods are not considered, for example the methods learning latent graphs: Semi-supervised classification with graph convolutional networks and Glomo: Unsupervised learning of transferable relational graphs.

5. The acknowledgement of the paper reveals location information, which may be a violation of anonymity.

Based on these cons, I think a more rigorous comparison is needed.

---

### Official Review · AnonReviewer2 · 2020-10-28
**A well-written paper, idea is simple, but experiment might be insufficient**

**Rating:** 5
**Confidence:** 4

**Review:**

==== Summary ====

This paper proposes a variant of Variational Autoencoders (VAEs) which takes extra topological information (e.g. adjacency matrix) into account in the loss function during training. The principle objective of the proposed GR-VAEs is to use the learned latent space features as the input for improvement on downstream tasks especially for classification.

==== Pros ====

+ This paper is well-written and the idea is clear and easy to follow.
+ The proposed GR-VAE, specifically the extra loss function on preserving the geodesic distance seems to be effective for learning desired latent space features from experiments.

==== Cons ====

My main concern is on the experiment on showing the improvement for downstream tasks from the embedding learned by GR-VAE.

-  I think the goal of the experiment is to demonstrate that, the embedding learned by GR-VAE is superior comparing to other kinds of features such as from the vanilla VAE, so I expect the embedding by GR-VAE is applied to different models (e.g. GraphSAGE, GCN, DeepWalk) instead of just DGI (at least I only notice that DGI is adapted), and on each model the result from using GR-VAE can outperform others using raw data features or other kinds of finetuned features. I think showing the improvement on different models can greatly enhance the soundness of this paper.

- In addition, for the experiment on chemical reaction I think there should be another baseline showing the result of DGI by using the raw data features if possible, this can further demonstrate the importance of GR-VAE. Also, for the experiment on text representation I also expect some result like using finetuned GR-VAE as the input for DGI in the chemical reaction experiment, currently the text representation experiment only shows the strength of the DGI itself, which does not make sense to me.

==== Reason for scoring ====

Overall, I think the proposed GR-VAE is sound if its strength can be demonstrated by more experiment mentioned above, and I am willing to upgrade my rating and vote to accept if such concern can be addressed during the rebuttal period.

==== Minor Comments ====

- The plots in Figure 3 is too blurry to distinguish between the cross marker and round marker.
- I notice for the synthetic dataset the direction of edges for each node is used as part of the input features, so what is the definition for the edge direction? Also, if we directly combine the raw data feature with embedding by some manifold learning technique, and input it into the vanilla VAE, can we get similar result (the graph topology is preserved) as GR-VAE has?

---

### Official Review · AnonReviewer4 · 2020-10-28
**Review for On the Importance of Looking at the Manifold: Reject**

**Rating:** 3
**Confidence:** 4

**Review:**

Summary:

The authors present a regularisation term for Variational Auotencoders that forces the distance of mapped points in the embedding space to be similar to the distance of those points in the metagraph of the data derived from relational information about these points. The intention of the regularisation term is to enforce a consistent graph between the original representation and the embedded representation in a manner that is agnostic to the structural choices of the model to be estimated.

Strengths:

1) The paper provides a good organisation of existing methods for utilising topological information, and, thus, positions its contributions well in relation to existing work.

2) The empirical results are presented honestly, even when they do not support the proposed method.

Weaknesses:

Ordered form less to more specific:

1) The intention of the paper is unclear.
	Is this intended as a review paper of recent research on graph neural networks or to present a new regularisation term? The title and abstract seem to imply that this is intended as a review, but the paper only considers three existing methods and a single modification to VAEs. Unfortunately, the paper is not convincing in either regard, and the idea of analysing topological information to improve classifications and representations is already well discussed in the literature and a very active area of ongoing research.

2)  The efficacy of the proposed regularisation is not convincingly supported either theoretically or empirically.
	The authors state, ‘Notably, GR-VAE is devised to infer topological information solely from a soft constraint, without any architectural requirements such as graph convolutions’ (line 1, p. 4), but this is not discussed further. The intention of graph convolutions is to explicitly encode assumptions about the relationships present in the data, in this situation why would I prefer a soft constraint to a well motivated, explicit one? If structural constraints were unduly restricting the expressiveness of the models, I would expect to see this borne out in the empirical results, but this is not the case across the chemical reaction and citation network experiments.

3) The paper struggles with clarity at points.
	Specifically, equation (1), which describes the regularisation term is unclear as written: does the plus sign in the exponent denote absolute value or something else? This notation is non-standard and I would not be able to faithfully recreate the results as written.
	Additionally, the method for constructing the meta-graph G should be discussed in more detail. From what is this graph derived? Is it an existing observed graph that describes the observed relationships between the data, ex. the citation network or pixels in an image, or does it describe adjacency of the observations as defined by the observed labels or other meta-information. My concern is that using a soft-constraint which effectively focusses the model on the labels in an ‘easy’ task such as MNIST classification or the synthetic data task hides the fact that the constraint is too soft to produce a useful regularisation of the model, as evidenced by the failure of GR-VAE relative to DGI or even vanilla VAE in the citation network and chemical reactions tasks.

Reasons for score:

I vote for rejecting the paper, as while I really do appreciate that the results of the paper are presented honestly, I think there are concerns with the current draft.

Questions for the rebuttal period:

Please refer to the questions in the weaknesses section.

---

### Official Review · AnonReviewer1 · 2020-11-02
**The idea is interesting but the method and experiments are not convincing**

**Rating:** 4
**Confidence:** 3

**Review:**

**Summary**:
This paper investigates different ways of incorporating topological information about the data in the machine learning models. The paper introduces a novel loss that aims to enforce the relational information between data points into the embedding space learned by a Vae on the node features. The experiments demonstrate that for data with a certain topology type, the introduced loss can provide performance when used together with existing methods. The paper opens up possibilities of further investigation into incorporating topological information (if available) into the learning procedure.

**Pros**:
1. The paper is very well-written and easy to follow. The illustrations also present the idea clearly.
2. The problem of understanding the importance of topological information is interesting, and could lead to future works.

**Cons**:
1. I am not sure if there is enough novelty for acceptance to ICLR, especially when the proposed method does not provide obvious benefit over existing methods, both in theory and practice. Specifically, the GR-VAE is a simple extension of VAE which does not yield good performance, unless it’s combined with DGI. Even in that case, it is not obvious to me the benefit is significant except in one case (namely for text representations), but when combined with DGI, it becomes unclear whether the performance boost is actually coming from the proposed loss or some other unintended regularization effect since DGI already uses message passing to incorporate the (local) topological structure. Am I missing something here?
2. Since the paper is positioned to be an experimental study, it is perhaps acceptable to have limited novelty or improvement. However, the findings in the papers are somewhat expected. For example, we already know that GNN [1] based methods are superior to other methods on citation benchmarks since they account for both features and topology. In this light, I feel there is not too much new insight in the paper to warrant a publication at ICLR.
3. For an empirical study, considering only four models may not be enough (e.g. different GNNs / graph models encode different topological information and that should be taken into account for a full spectrum). The same holds for feature-based methods. Some examples of this are Deep walk [3], GNN architecture or different VAE architecture and loss (especially there are so many variations of VAE). Most importantly, a comparison to [2] is missing.

**Comment**:
1. “Regularized” -> “regularizer” in conclusion line 6 paragraph 2?
2. For a double-blind reviewing process, the funding information should perhaps be removed, although the one in this paper does not expose the authors' identity.

**Conclusion**:
While the work has interesting motivations and is well-written, it has not done a convincing job at demonstrating the effectiveness of their proposed method or shown a thorough experimental analysis. As such, I am inclined to reject the paper in its current form.

**Reference**:

[1] Semi-Supervised Classification with Graph Convolutional Networks (https://arxiv.org/abs/1609.02907)

[2] Variational Graph Auto-Encoder (https://arxiv.org/abs/1611.07308)

[3] Deepwalk: Online learning of social representations (https://arxiv.org/abs/1403.6652)


**================================== Update after rebuttal ==================================**

It seems that the authors have only provided a general comment for all reviewers, which is understandable since most criticisms from all reviewers are on the same weaknesses of the paper.
While I appreciate the author's effort in adding more experiments, I do not think the added experiments and reply properly addressed the concerns shared by other reviewers and myself. For example, it is still unclear what the advantage of the proposed method is or what insights we could gain from this study.
I think this paper is not ready for publication. As today is the last day of discussion period, I will maintain my original assessment.

---

### Author Response · Authors · 2020-11-25
**Answer to the reviewers**

Answer to all reviewers.

First of all we thank all the reviewers for the time and effort dedicated to reading and reviewing our paper. Seeing that several cons were shared across reviewers we decided to address them together.

The intent of the paper was to study the influence of the topology across the scale shown in the paper. The addition of the GR-VAE is motivated by the intent of filling a particular shade of that spectrum.

We appreciate the recommendation of including more more models into the study. We added text baselines for GAE and GraphSAGE in the text study, and we are aware that the study could be further expanded with more models. Furthermore, we expanded the text classification results for the DGI using subsampled graphs (i.e. same sampling method by which GR-VAE is trained) and incorporating the representations of the variational autoencoders as features.

We want to make the same point about the datasets, it has been commented that downstream prediction tasks may not be the optimal comparison method. It was a direct way for us to asses the goodness of the representations in respect to a specific task, however we are aware that can in itself be limiting. We welcome the different suggestions done in the comments (e.g. controlling adjacency or corrupting the input graphs) and we will be looking at them.

Finally, just reiterate our thanks for the reviewers' comments and time invested in looking into our work. All the feedback we received is highly welcome and a helpful light on how to improve and iterate this work.

---

### Decision · Program_Chairs · 2021-01-07
**Final Decision**

**Decision:**

Reject

**Comment:**

While the motivation of the paper is interesting the reviewers expressed concerns about the experimental setup, comparison to related work, and paper framing. For experiments, it was unclear why authors compared such disparate methods instead of more fine-grained adjustments (e.g., such as corrupting graphs as suggested by R3). For comparison, other methods such as Deep Walk and VGAE (as suggested by R1) seemed missing. I think the biggest issue however was with framing: as the reviewers pointed out, it was not clear enough how looking at downstream performance relates to looking at the manifold. In fact the paper title is much too general and is also well-known: manifold learning has been around for 15+ years. I would urge the authors to take the recommendations of reviewers and either design new experiments that explicitly target the manifold or reframe the paper to design new evaluation metrics for latent (possibly structured) generative models.